

# The long-term effects of COVID-19 on pulmonary status and quality of life

Ayedh Alahmari[1], Gokul Krishna[1], Ann Mary Jose[1], Rowaida Qoutah[1],
Aya Hejazi[1], Hadeel Abumossabeh[1], Fatima Atef[1], Alhanouf Almutiri[1],
Mazen Homoud[2], Saleh Algarni[3,4], Mohammed AlAhmari[5,6],
Saeed Alghamdi[7], Tareq Alotaibi[3,4], Khalid Alwadeai[8],
Saad Alhammad[8] and Mushabbab Alahmari[9]

[1] Department of Respiratory Therapy, Batterjee Medical College, Jeddah, Saudi Arabia
[2] Department of Respiratory Therapy, King Abdulaziz University, Jeddah, Saudi Arabia
[3] Department of Respiratory Therapy, College of Applied Medical Sciences, King Saud bin
  Abdulaziz University for Health Sciences, Riyadh, Saudi Arabia
[4] King Abdullah International Medical Research Center, Riyadh, Saudi Arabia
[5] Dammam Medical Complex, Eastern Health Cluster, Dammam, Saudi Arabia
[6] Department of Respiratory Care, Prince Sultan Military College of Health Sciences, Dammam,
  Saudi Arabia
[7] Clinical Technology Department, Respiratory Care Program, Umm Al-Qura University,
  Makkah, Saudi Arabia
[8] Department of Rehabilitation Science, College of Applied Medical Sciences, King Saud
  University, Riyadh, Saudi Arabia
[9] Department of Respiratory Therapy, University of Bisha, Bisha, Saudi Arabia

Corresponding author
Ayedh Alahmari,
ayedh.dhafer@bmc.edu.sa

## ABSTRACT

**Background:** Few studies have looked at how SARS-CoV-2 affects pulmonary function, exercise capacity, and health-related quality of life over time. The purpose of this study was to evaluate these characteristics in post COVID-19 subjects 1 year after recovery.

**Methods:** The study included two groups. The case group included post COVID-19 subjects who had recovered after a year, and the control group included healthy participants who had never tested positive for COVID-19.

**Results:** The study screened 90 participants, 42 of whom met the eligibility criteria. The findings revealed that the majority of post COVID-19 subjects had relatively normal lung function 1-year post-recovery. A significant reduction in DLCO (B/P%) was observed in the case group *vs.* control. The exercise capacity test revealed a clinically significant difference in distance walked and a significant difference in the dyspnea post-walk test in the case group compared to the control group. The case group's health-related quality of life domain scores were significantly affected in terms of energy/fatigue, general health, and physical function.

**Conclusions:** The post COVID-19 subjects were shown to have well-preserved lung function after 1 year. However, some degree of impairment in diffusion capacity, exercise capacity, and health-related quality of life remained.

## INTRODUCTION

In December 2019, China reported a novel coronavirus, known as severe acute respiratory syndrome coronavirus 2 (SARS-CoV-2) (*Du Toit, 2020*). The World Health Organization (WHO) issued an emergency alert regarding the coronavirus on January 30, 2020, and later declared it a pandemic. COVID-19 is regarded as one of the most serious infectious diseases of the twenty-first century, with WHO reporting 6,972,152 deaths as of 7[th] October 2023 (*World Health Organization, 2023*).

The coronavirus enters the human cell *via* respiratory droplets by binding to angiotensin-converting enzyme (ACE2) receptors (*Du Toit, 2020*). Once the virus enters the cell, an inflammatory response occurs, resulting in fluid leakage, increased lymphocyte production, and the release of antiviral cytokines into the alveolar septa and interstitial compartments. These effects then cause acute respiratory failure or acute respiratory distress syndrome (ARDS). COVID-19 has the potential to affect all body systems. The most frequently reported symptoms of COVID-19 are high-grade fever, cough with expectoration, and dyspnea (*Ali, Sadiq & Yunus, 2021*; *Cabrera Martimbianco et al., 2021*; *Piazza et al., 2020*; *Rubino et al., 2020*; *Tian et al., 2020*; *Verdecchia et al., 2020*; *Wild et al., 2022*).

Smoking, disease comorbidities, age, female sex, respiratory diseases, psychological conditions, and hospitalizations have all been shown to negatively impact the severity and recovery of COVID-19 (*Hastie et al., 2022*). Post-COVID-19 symptoms can last for days, weeks, or are known by a variety of names, including post-COVID-19 conditions, long COVID, and post-acute COVID-19 syndrome (*Nalbandian et al., 2021*). Long term COVID-19 complications were defined as a post COVID-19 condition (PCC) by the WHO Delphi consensus that occurs "in individuals with a history of probable or confirmed SARS- CoV infection, usually 3 months from the onset of COVID-19 with symptoms that lasts for at least 2 months and cannot be explained by an alternative diagnosis" (*World Health Organization, 2021a*, *2021b*). PCC may affect multiple organ systems (*Thaweethai et al., 2023*). The individuals who experience persistent respiratory symptoms after COVID-19 are likely to have multiple and diverse causes, which may also coexist, depending on their pre-existing medical conditions, the severity of their acute COVID-19 illness, the duration of their ICU stay, and the length of their hospital stay (*Achkar, Jamal & Chaaban, 2022*). Post-COVID-19 conditions may also have economic and societal consequences; the COVID-19 Longhauler Advocacy Project found that 44% of patients with long COVID symptoms were unable to work with comparable quality and capacity as before COVID-19 (*Project, 2022*).

The pathophysiology, acute complications, prevention, and initial management of COVID-19 have all been extensively studied; however, the long-term consequences are less well understood (*Nalbandian et al., 2021*). Regardless of the initial severity, one in every 10 people infected by the virus has persistent symptoms after 4 weeks (*Kostev et al., 2022*). Impaired concentration, generalized anxiety disorder, fatigue, muscle weakness, and functional mobility disorders are all reported symptoms of PCC (*Groff et al., 2021*).

The earlier SARS-CoV and Middle East respiratory syndrome coronavirus (MERS-CoV) follow-up studies revealed impairments in lung function and quality of life that persisted for months to years after recovery from the acute illness (*Ahmed et al., 2020*). Only a few studies have reported the long-term effects of COVID-19 on lung function. A 3-months follow-up study found that COVID-19 pneumonia survivors had short-term radiological abnormalities (*Zhao et al., 2020*). Whereas, post 1 year study reported the abnormalities regressed and not shown any progression (*Vijayakumar et al., 2022*). The results of almost all radiological and lung function studies that were conducted at the time of discharge revealed decreased or abnormal findings (*Liang et al., 2020*; *Soriano et al., 2022*).

This observational study aimed to assess lung function, exercise capacity, and health-related quality of life in post COVID-19 subjects after 1 year.

## MATERIALS AND METHODS

### Participants and design of the study

We conducted a descriptive, case-controlled study using a matched sampling technique based on age, gender, and body mass index (BMI). The case group included post COVID-19 subjects who were 1-year post-recovery; the control group included healthy subjects who had never tested positive for COVID-19. Participants in both groups were enrolled from the community and were identified from the Saudi government's contact tracing system over the course of 1 year. Participants in the case group required a positive real-time polymerase chain reaction (RT-PCR) test 1 year before the study and negative tests thereafter. Along with the positive test, participants must have had COVID-19 symptoms including fever, loss of taste and smell, coughing, or body ache. The participants who met the inclusion criteria for the control group were included as long as they had no known history of SARS-CoV-2 infection as confirmed by the Saudi government's contact tracing system. Subjects with a history of chronic obstructive pulmonary disease, asthma, interstitial lung disease, tuberculosis, congestive heart failure, cardiovascular diseases, smokers, and those who could not perform spirometry correctly were excluded from both groups.

This study was approved by the institutional ethical committee at Batterjee Medical College (BMC) (RES 2022-42). The purpose of the study was explained to all participants, and both groups signed an informed consent form.

### Assessments

Demographics and medical history were collected from each participant, followed by forced vital capacity, diffusion lung capacity maneuvers (DLCO), and a 6 min walk test (6MWT). Participants were also asked to complete the general health survey (SF-36) questionnaire to measure health-related quality of life (HRQoL).

### Pulmonary function testing

A certified respiratory therapist performed the pulmonary function test (PFT) at the BMC respiratory therapy department (RT). A medical doctor was present during the study to

manage any emergency that might occur. The PFT was performed following the American Thoracic Society (ATS) and European Respiratory Society (ERS) guidelines (*Stanojevic et al., 2022*). Spirometry and diffusion capacity were measured using master screen PFT by trained RTs (powered by Sentry Suite).

Forced vital capacity (FVC), forced expiratory volume in the first second (FEV1), FEV1/FVC, and DLCO were all measured using a single-breath technique with hemoglobin concentration adjustment. FVC (B/P), FEV1 (B/P%), and DLCO (B/P%) were also measured. B/P% is defined as the ratio of the best value produced by the participant and the predicted value of the participant measured according to age, height, and weight. All participants performed the procedure three times; we recorded and used the best value in our analysis. All PFT measurements were expressed as absolute values and as a percentage of predicted normal values (% predicted).

### 6-min walk test

Participants in both groups completed the 6 min walk test (6MWT) after completing the PFT to assess functional capacity and physical activity following the ATS guidelines. The 6MWT is a valid and reliable test to measure maximal exercise performance and physical activity. The test was performed using a 30-m corridor in case of an emergency. A crash cart was prepared near the corridor. Participants were instructed to dress comfortably and to wear appropriate walking shoes. Participants' vital signs were recorded before and after completing 6MWT. The Borg scale was used to record the intensity of breathlessness and fatigue. The number of laps completed by the participants were recorded using a lap counter. The test was terminated if chest pain, intolerable dyspnea, leg cramps, staggering, or diaphoresis occurred (*Holland et al., 2014*).

### Health-related quality of life

We assessed the health-related quality of life (HRQoL) using the SF-36 survey questionnaire (*Ware & Sherbourne, 1992*). The SF-36 questionnaire is a subset of questions from longer instruments used in medical outcome studies. The version of SF-36 used in this study is specifically known as RAND SF-36. This questionnaire was developed at RAND as part of their Medical Outcomes Study (*Rand Corporation, 2023*). A 36-item patient-reported questionnaire with eight health domains was included in the survey. These domains include: physical functioning (10 items), social functioning (two items), role limitation due to emotional problems (four items), role limitation due to physical health problems (four items), bodily pain (two items), general health perceptions (five items), emotional well-being (five items), and energy/fatigue (five items and four items, respectively). Scores for each domain ranged from zero (worst) to 100 (best), with a higher score defining a more favorable healthy state.

### Statistical analysis

SPSS windows v.25 was used for data analysis (Chicago, IL, USA). To determine normality, we used the Shapiro–Wilkins test due to the small sample size. Continuous variables following normal distribution were represented as mean ± standard deviation and median ± interquartile range for non-normally distributed variables. Categorical variables were

expressed in number and percentage. For continuous variables, an independent sample t-test and a Mann-Whitney U test were used. A $p$ value of < 0.05 was considered statistically significant.

## RESULTS

### Demographic data

We screened 90 participants for the study, 42 of whom met the eligibility criteria. We excluded two people from the case study and one from the control group because they could not perform the PFT correctly. There were 19 participants in the case group and 20 participants in the control group for the final analysis (Fig. 1). There was no significant difference between the groups in regards to age, gender, and BMI. The most commonly reported COVID-19 symptoms in the case group were fever, loss of smell, and taste. All of the participants were free of COVID-19-associated symptoms at the time of the study. The demographics of the participants are shown in Table 1.

### Comparison of spirometry and diffusion capacity between case and control groups

After 1 year, COVID-19 patients in the case group showed no significant difference in spirometry measurements from those in the control group (Table 2). The FVC, FEV1, and FEV1/FVC (best/predicted percentage) were all within the normal range. The absolute values of carbon monoxide diffusion capacity did not differ statistically between the case group and the control group. However, there was a statistically significant difference in DLCO (B/P%) between the control and case groups, with a $p$ value of 0.014 (Table 3).

### Exercise capacity (6-MWT)

The 6-MWT revealed a reduction in the total distance walked in the case group compared to the control group W. However, the difference in walking distance between the groups was not statistically significant. In the post walk test, the case group had significantly more dyspnea than the control group (Table 4).

### Comparison of health-related quality of life between post-COVID-19 and control group

The SF 36 questionnaire was used to explore the HRQoL among the case and control groups. In almost all domains of HRQoL, the post-COVID-19 case group participants showed lower scores compared to the control group, with some domains significantly more affected than others. When compared to the control groups, there was a significant reduction in three domains, including energy/fatigue 56.32 ± 17.93 *vs*. 66 ± 10.58 ($p$-value = 0.046); general health 70.53 ± 14.32 *vs*. 82.25 ± 15.43 ($p$-value, = 0.019); and physical functioning 80 (46.25) *vs*. 95 (18.75) ($p$-value = 0.010). Even though the physical functioning domain was significantly affected, the role limitation associated with physical factors was not significant. Role limitation associated with emotional problems was the least affected domain. Table 5 represents the SF-36 domain outcomes of both groups. Figure 2 shows the average SF-36 questionnaire score for the case and control groups.

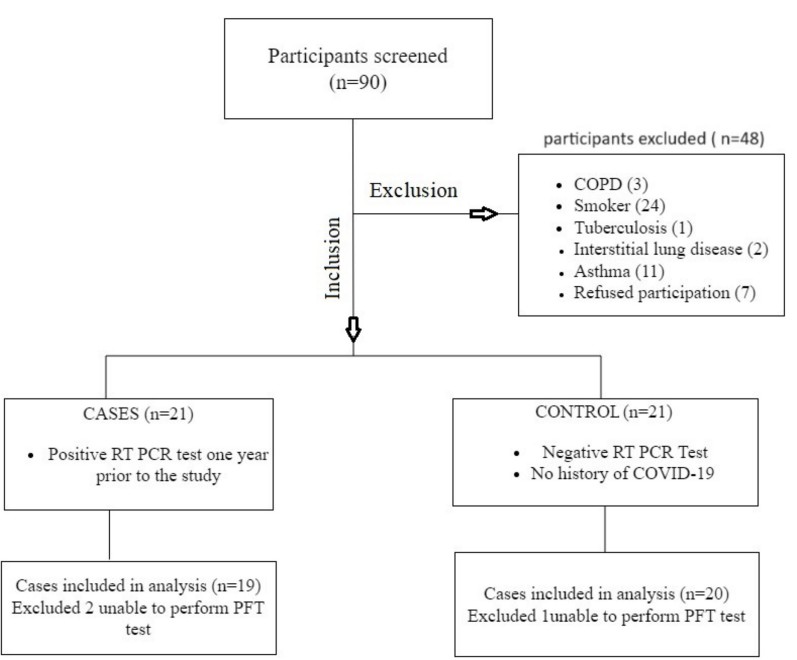

**Figure 1 Flowchart of data collection.**

**Table 1 Demographic and clinical characteristics of participants.**

| Variable | Case group (*n* = 19) | Control group (*n* = 20) | *p* value |
|---|---|---|---|
| Gender *n* (M/F) | 6/13 | 7/13 | 0.13 |
| Age (mean ± SD) | 32 ± 12.8 | 31 ± 12 | 0.27 |
| BMI (kg/m²) (mean ± SD) | 26 ± 6.4 | 26 ± 6.3 | 0.31 |
| Hospital admission due to COVID-19 *n* (%) | 3 (15) | 0 | – |

**Table 2 Spirometry comparison among the case and control groups.**

| Outcome (Mean ± SD) | Case group (*n* = 19) | Control group (*n* = 20) | *p* value |
|---|---|---|---|
| FVC (L) | 3.91 ± 0.92 | 3.84 ± 0.86 | 0.573 |
| FVC (B/P %) | 101.37 ± 14.32 | 105 ± 15.50 | 0.484 |
| FEV1 (L) | 3.18 ± 0.68 | 3.20 ± 0.66 | 0.828 |
| FEV1 (B/P %) | 98.84 ± 6.727 | 100.70 ± 8.405 | 0.636 |
| FEV1/FVC | 82.16 ± 7.73 | 83.20 ± 7.79 | 0.852 |

**Table 3 The comparison of diffusion capacity among the case and control groups.**

| Outcome (Mean ± SD) | Case group (*n* = 19) | Control group (*n* = 20) | *p* value |
|---|---|---|---|
| DLCO (mmol/kPa.min) | 19.94 ± 7.08 | 20.15 ± 3.90 | 0.569 |
| DLCO (B/P %) | 77.58 ± 13.35 | 97.42 ± 16.59 | 0.014 |

**Table 4 A comparison of 6MWT among the case and control groups.**

| Outcome | Case group (n = 19) | Control group (n = 20) | p value |
|---|---|---|---|
| Distance walked (mean ± SD) | 495.63 ± 72.4 | 521.45 ± 65.6 | 0.569 |
| Borg Scale Dyspnea-post walk Median (IQR) | 2 (4.5) | 0.5 (1) | 0.01 |
| Borg scale fatigue–post walk Median (IQR) | 1 (2.75) | 0.5 (2) | 0.184 |
| SpO2%-post walk (mean ± SD) | 98.10 ± 0.81 | 98.91 ± 0.90 | 0.47 |

**Table 5 A comparison of the SF36 domain outcomes between the case and control groups.**

| Outcome | Case group (n = 19) Mean ± SD | Control group (n = 20) Mean ± SD | p value |
|---|---|---|---|
| Energy fatigue | 56.32 ± 17.93 | 66 ± 10.58 | 0.046 |
| Emotional well-being | 56.63 ± 20.49 | 68.8 ± 19.27 | 0.064 |
| General health | 70.53 ± 14.32 | 82.25 ± 15.43 | 0.019 |
| **Outcome** | **Case group (n = 19) median (IQR)** | **Control group (n = 20) median (IQR)** | **p value** |
| Physical functioning | 80 (46.25) | 95 (18.75) | 0.010 |
| Role limitation-physical | 75 (25) | 100 (25) | 0.258 |
| Role limitation-emotional | 100 (67) | 100 (33) | 0.428 |
| Social functioning | 75 (38) | 87.5 (34) | 0.127 |
| Pain | 88 (12) | 90 (22.75) | 0.184 |
| Health change | 50 (25) | 50 (25) | 0.607 |

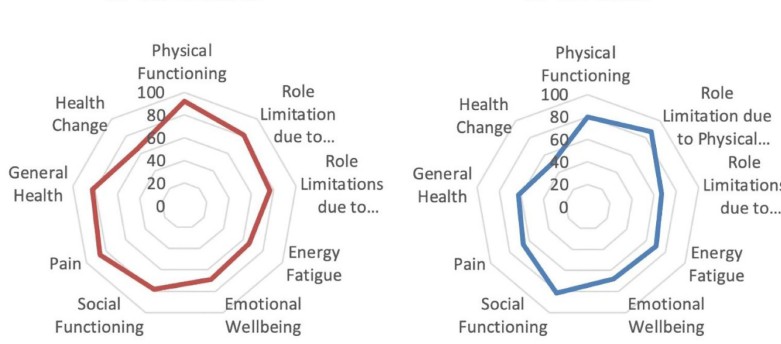

**Figure 2 Average score for control and case group SF-36 questionnaire.**

# DISCUSSION

In this study, we found that all post COVID-19 subjects were symptom-free after 1 year and had relatively normal and well-preserved lung function. Although the diffusion capacity was within the normal range, we observed a significant reduction in DLCO (B/P

%). A reduction in exercise capacity and HRQoL was observed in the case subjects, with the post-dyspnea score, energy/fatigue, physical functioning, and general health significantly affected.

Dyspnea was one of the most common post-acute COVID-19 persisting symptoms with prevalence ranging from 42–66% in 3 to 6 month follow-up studies (*Chopra et al., 2021*; *Garrigues et al., 2020*; *Halpin et al., 2021*). This is similar to the findings of the study done by *Havervall et al. (2021)* where they followed up on post-COVID patients for 8 months. Patients who still tested positive with no persistent symptoms 8 months after the onset of COVID-19 had more severe dyspnea and fatigue than patients who tested negative 8 months after the onset of COVID-19 (*Havervall et al., 2021*). The average age of our post COVID-19 participants was 32 and only 15 percent required hospitalization compared to other studies where participants were much older and had higher percentage of hospitalization which could be a possible reason for absence of persistent symptoms in the current study.

The anomalies of pulmonary ventilation and diffusion function were the most reported physiological impairment with COVID-19 (*Daher et al., 2020*). Studies have found some degree of impairment in lung function and a significant reduction in diffusion capacity from the date of medical discharge up to 8 months after the acute illness (*Ahmed et al., 2020*; *Huang et al., 2020*; *Lehmann et al., 2022*; *Mendez et al., 2021*; *Zhao et al., 2020*). *Zhang et al. (2021)* discovered that severe COVID-19 cases had significantly more DLCO impairment and lower lung capacities than non-severe cases. In contrast to previous studies that found a higher percentage of restrictive lung impairment, our results found normal or near normal FVC, FEV1, and FEV1/FVC ratios after 1 year. However, we also observed a significant difference in DLCO (B/P%) with normal diffusion capacity.

Another pulmonary sequela observed in post-acute COVID-19 patients was a decrease in cardiopulmonary capacity, which is why patients were assessed using 6MWT (*Soriano et al., 2022*). A total of 6MWT results have been shown to have a moderate relationship with pulmonary function test results and HRQoL (*Holland et al., 2014*). The prevalence of muscle weakness and decreased walking distance in COVID-19 patients was comparable to that observed in previous SARS and MERS survivors (*Ahmed et al., 2020*). In a 6-month follow-up study of 1,733 Chinese patients, *Huang et al. (2021)* found a decrease in median 6 min walking distance compared to the normal reference range in survivors, whereas the follow-up study by *Wu et al. (2021)* reported progressive improvement in walking distance from 535 m at 3 months to 615 m at 1 year. In our study we did not find a significant difference between the groups. The post walk test dyspnea was significantly higher among post-COVID-19 subjects.

According to a recent systematic review by *Figueiredo et al. (2022)*, HRQoL impairment persisted in both mental and physical domains in post-COVID-19 patients months after discharge. A similar finding was observed by *Nandasena et al. (2022)* and reported the need for implementing programs to improve quality of life in post COVID-19 patients. An extended period of isolation, intense media attention, phobia and rejection from the general public (especially in the early stages), and fear of disease transmission are all potential causes of mental health problems (*Hui et al., 2005*). Also, fatigue is one of the

most common symptoms reported in COVID-19 at 6 months and above, in systematic review fatigue was found in 30.94% of the patient in 6–12 months and in 34.5% after 12 months (*Ma et al., 2022*). Survivors of acute lung injury (ALI) and ARDS have reported impairment of HRQoL at 1 to 5 years after recovery (*Thaweethai et al., 2023*), we discovered a significant decrease in the HRQoL in post COVID-19 subjects' when compared to a healthy population, with major effects on energy/fatigue, general health, and physical function. Our research revealed a significant difference in physical functioning but no significant differences in the role limitations studied. This may be due to the fact that physical functioning items measure the physical limitations of daily activities, including lifting and carrying groceries, climbing up the stairs, bending, kneeling, *etc*. Meanwhile, physical role limitations mainly measure work limitations, time spent at work, and difficulties in performing work (*Ware et al., 1993*). The majority of participants were without jobs, which may explain the discrepancy of the results between physical functioning and the physical limitations.

According to *Palakshappa et al. (2021)*, ARDS survivors recovered their lung capacity 6–12 months after hospital discharge, but suffered from long term physical, cognitive, and mental impairments. Since there is increase in ARDS incidence with COVID-19, it is important to re-evaluate the severity and lasting of post COVID 19 symptoms for an additional year. We also suggest evaluating the severity of impaired lung function based on ICU admission and the duration of COVID-19.

Our study has several limitations, including a small sample size, the absence of baseline lung function and diffusion capacity tests, and the lack of information regarding outcomes based on severity and hospital admissions information. However, we tried to overcome these limitations by recruiting a matched control group. Future research is needed to look into the long-term effects of COVID-19 in both critical and non-critical cases.

## CONCLUSIONS

Post COVID-19 subjects were shown to have well-preserved lung function 1 year after infection. However, there was some impairment in lung diffusion capacity, exercise capacity, and HRQoL. Further studies need to be conducted in order to determine the long-term effects of post-COVID-19 symptoms, 2 years after the conclusion of the current study, to evaluate the lasting effects of post COVID-19 symptoms.

### Funding
The authors received no funding for this work.

### Competing Interests
The authors declare that they have no competing interests.

### Author Contributions
- Ayedh Alahmari conceived and designed the experiments, analyzed the data, authored or reviewed drafts of the article, and approved the final draft.

- Gokul Krishna conceived and designed the experiments, analyzed the data, prepared figures and/or tables, and approved the final draft.
- Ann Mary Jose analyzed the data, prepared figures and/or tables, and approved the final draft.
- Rowaida Qoutah conceived and designed the experiments, authored or reviewed drafts of the article, and approved the final draft.
- Aya Hejazi performed the experiments, prepared figures and/or tables, and approved the final draft.
- Hadeel Abumossabeh performed the experiments, prepared figures and/or tables, and approved the final draft.
- Fatima Atef performed the experiments, prepared figures and/or tables, and approved the final draft.
- Alhanouf Almutiri performed the experiments, prepared figures and/or tables, and approved the final draft.
- Mazen Homoud analyzed the data, prepared figures and/or tables, and approved the final draft.
- Saleh Algarni conceived and designed the experiments, authored or reviewed drafts of the article, and approved the final draft.
- Mohammed AlAhmari performed the experiments, authored or reviewed drafts of the article, and approved the final draft.
- Saeed Alghamdi performed the experiments, authored or reviewed drafts of the article, and approved the final draft.
- Tareq Alotaibi performed the experiments, authored or reviewed drafts of the article, and approved the final draft.
- Khalid Alwadeai performed the experiments, authored or reviewed drafts of the article, and approved the final draft.
- Saad Alhammad performed the experiments, authored or reviewed drafts of the article, and approved the final draft.
- Mushabbab Alahmari performed the experiments, authored or reviewed drafts of the article, and approved the final draft.

## Human Ethics

The following information was supplied relating to ethical approvals (*i.e.*, approving body and any reference numbers):

Batterjee medical college granted ethical approval to carry out the study within its facilities (ethical approval number: RES 2022-42).

## Data Availability

The raw data is available in the Supplemental File.

## Supplemental Information

Supplemental information for this article can be found online at http://dx.doi.org/10.7717/peerj.16694#supplemental-information.

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
