# Peer review of "The long-term effects of COVID-19 on pulmonary status and quality of life"

_PeerJ, doi:10.7717/peerj.16694_

## Round 0.1 · original submission · Major Revisions

The authors evaluated long-term effects of COVID-19 on pulmonary status and
quality of life. However, in addition to the reviewers comments, I have also some - comments that bothered me:

1. The main concern is related to the methodolgy: the most important missing data of this study is the clinical signs of COVID-19 of the cases. In fact, the sequelae of The diseases are mainly related to its severity and its duration, so using just the results of the PCR is not really sufficient. One can be positive for PCR but he has no symptoms, and vice-versa.
Also, the selection of the control group is not well defined.

2. The introduction needs to be reorganized (especially in lines 99-115 where the ideas are repeated which makes them confused. also, the hypothesis needs to be defined.
Line 82: update the data, please
Line 90: delete, please
Line 94: The
Line 96: delete had
Line 113: infected may be affected (infected by the virus)
Line 116: provide the references for these studies

3. The discussion is very simplistic and needs some revision. It contains also multiple "orphan" and "unclear data"( exp: line 246: "whereas in our study the majority of participants were symptom-free". when were participants symptoms-free?
Line 248-248: what do you mean here? provide more details please
Line 270: decrease in score? of what?
Line 274: ARDS, ALI???
Line 279: be evaluated (will), COVID (COVID-19)
Line 279-282: why did you insist on 5 years? why not 3? 7?...? is there scientific evidence?
allergies ???? revise all these parts please

I recommend avoiding using COVID-19 survivors since we don't know exactly the severity of the disease.

The conclusion is very simplistic and did not provide interesting information.
Revise, please.

Write p-value (not P)

At last, your table should be improved for their quality, make all factors in one line, please.

·

Basic reporting

This article is clear and concise. The authors describe changes of lung function, HR-QOL and exercise capacity 1 year after covid-19 infection of patients in Saudi Arabia

Experimental design

1. Can the authors explain the choice of utilizing SF-36 questionnaire to explore HR-QOL in covid-19 survivors. Has this questionnaire been validated for use in covid-19 survivors.
2. Are authors aware of the severity of covid-19 illness in the patients within the case group?
3. The exclusion criteria included patients with prior lung diseases. Did the authors account for patients with heart disease or pulmonary hypertension? DLCO reduction was noted to be significant in the case group vs control group. Pulmonary hypertension and heart disease may also lead to similar observation.
4. Can the authors explain the discrepancy observed between the reduction in physical function and role limitation?

Validity of the findings

1. Can the authors provide some clarification on the significance of this study in comparison to the recently published RECOVER trial in JAMA (https://jamanetwork.com/journals/jama/fullarticle/2805540).

Additional comments

N/A

Reviewer 2 ·

Basic reporting

Thank you for the opportunity to review this interesting work.

Overall, I would like to suggest that the article will be reviewed for clarity and inaccuracies. Please see some specific comments below:
• “The control group included healthy subjects who had never tested positive for COVID-19”. How often these subjects were tested to confirm that they have not had COVID rather than being asymptomatic?
• Despite the above, the authors mentioned “the case group most commonly reported COVID-19 symptoms were fever, loss of smell, and test” (line 204). I assume this statement refers to the acute stage, but it's not clear from the text.
• Please explain the meaning of DLCO (B/P%) the first time it is mentioned.
• Information in lines 218-222 is not clear: “reaching a clinically significant difference of …However, the difference in walk distance between the groups was not statistically significant.” I suggest differentiating and indicating clearly when referring to the previous literature or the current study.
• Please explain how was the borg test used. I am curious of why the authors reported score for dyspnea and fatigue separately. Also, it seems inaccurate to state, “the Borg scale was used to record baseline dyspnea and fatigue as well as baseline blood pressure, SPO2, and heart rate” (line 173-174). In the borg scale: “Participants are asked to rate their exertion on the scale during the activity, combining all sensations and feelings of physical stress and fatigue.” (https://doi.org/10.1093/occmed/kqx063). The use and reporting of this scale should be clarified throughout the manuscript.
• p-values should be added to the difference reported (line 229-231).
• Information from lines 205 vs 239 and 246 contradicted regarding the presence of COVID symptoms.
• Please clarify in the manuscript what do you mean by “seropositive” positive or negative patients.
• You should consider using “post 6MWT dyspnea” instead of “the walk test dyspnea.”
• Line 271. How many months?
• SF-36 “/” is missing in energy/fatigue.
• Why “for another 5 years” instead of 2 or 10? (Line 279)

Experimental design

This study addresses a knowledge gap in a segment of the post-COVID population (post-COVID survivors without persistent symptoms), and this needs to be clarified in the manuscript.
Ethical approval was obtained for this study.

Validity of the findings

My main comment is regarding the generalizability of the findings to this study to the post COVID population, since: 1) “all the participants were free of COVID-19 associated symptoms” (line 205) which indicates that they don’t have long COVID, and 2) subjects with preexisting comorbidities (line 139) were excluded from the study. The literature indicates that pre-existing comorbidities contribute to disease severity during infection and increase the likelihood of long COVID.

---

## Round 0.2 · Minor Revisions

The authoues should updates some of their references.
Line 81: Delete ("Organisation 2019)".
Try to improve the quality of the tables.

·

Basic reporting

The authors have addressed all my questions and concerns. Thank you

Experimental design

The authors have addressed all my questions and concerns. Thank you

Validity of the findings

The authors have addressed all my questions and concerns. Thank you

Additional comments

The authors have addressed all my questions and concerns. Thank you

Reviewer 3 ·

Basic reporting

The authors use many references from 2020 and 2021, when the pandemic just began. Much evidence has emerged in recent months of persistent symptoms, especially in good quality systematic reviews. I think authors should update their references.

Experimental design

It's basic. But enough for this topic.

Validity of the findings

The authors should compare with similar series, that is, similar age, use of MV or hospitalization rate. The big problem I see is that they had only 15% of hospitalized patients, who are the ones who mainly have the consequences studied.

Additional comments

Dear authors:

Some comments that must be addressed:

Major comments:

The authors report in the results that the post-COVID group had a clinical difference with the control group in the 6MWT given that there was a difference greater than 25.82 meters. However, this difference is established as a change in a subject with respect to an intervention (improvement or deterioration). Therefore, this value should be used within subjects and not to compare different subjects. This analysis should not be discussed either in results or in discussion.

Only 15% of the subjects were hospitalized. Although persistent symptoms may be present in all subjects who have suffered from Covid-19, they are more likely to be suffered by older adults, hospitalized for more than 10 days, on mechanical ventilation, etc. This should be highlighted throughout the text. On the other hand, when compared with other studies, it is important to highlight whether the populations have similar characteristics, especially in age and hospitalization rates.

The authors use many references from 2020 and 2021 when we were still learning about this disease. There is much of this data updated with longitudinal studies with larger sample sizes or reviews with meta-analysis with more powerful conclusions. I suggest authors update the references for symptoms, radiological abnormalities, quality of life, physical capacity by eliminating those from 2020 and using 2022 or 2023.

Minor comments:
Affiliation 1 has respiratory therapy with the first letters in lowercase. However, in 2 onwards the first letters are capitalized.
line 103-105: The data presented does not belong to Cutler 2022. Cutler reports this data based on another study. Please cite the primary reference

Update the 6MWT recommendations. The authors present the one from 2002, but it was updated in 2014.: Holland AE, Spruit MA, Troosters T, et al. An official European Respiratory Society/American Thoracic Society technical standard: field walking tests in chronic respiratory disease. Eur Respir J. 2014;44(6):1428-1446. doi:10.1183/09031936.00150314

---

## Round 0.3 · accepted · Accept

The authors have answered all reviewer comments.

Reviewer 3 ·

Basic reporting

The authors responded to all my comments satisfactorily.

Experimental design

The authors responded to all my comments satisfactorily.

Validity of the findings

The authors responded to all my comments satisfactorily.

Additional comments

The authors responded to all my comments satisfactorily.